# Spatial distribution and determinants of measles vaccination dropout among under-five children in Ethiopia: A spatial and multilevel analysis of 2019 Ethiopian demographic and health survey

**Alebachew Ferede Zegeye**[ORCID][1]*, **Enyew Getaneh Mekonen**[2], **Berhan Tekeba**[3], **Tewodros Getaneh Alemu**[3], **Mohammed Seid Ali**[3], **Almaz Tefera Gonete**[3], **Alemneh Tadesse Kassie**[4], **Belayneh Shetie Workneh**[5], **Tadesse Tarik Tamir**[3], **Mulugeta Wassie**[6]

1 Department of Medical Nursing, School of Nursing, College of Medicine and Health Sciences, University of Gondar, Gondar, Ethiopia, 2 Department of Surgical Nursing, College of Medicine and Health Sciences, University of Gondar, Gondar, Ethiopia, 3 Department of Pediatrics and Child Health Nursing, School of Nursing, College of Medicine and Health Sciences, University of Gondar, Gondar, Ethiopia, 4 Department of Clinical Midwifery, School of Midwifery, College of Medicine and Health Sciences University of Gondar, Gondar, Ethiopia, 5 Department of Emergency and Critical Care Nursing, School of Nursing, College of Medicine and Health Sciences, University of Gondar, Gondar, Ethiopia, 6 School of Nursing, College of Medicine and Health Sciences, University of Gondar, Gondar, Ethiopia

* alexferede24@gmail.com

## Abstract

### Background

Each year, vaccine-preventable diseases cost the lives of 8.8 million under-five children. Although vaccination prevents 1–2 million childhood deaths worldwide, measles vaccination dropouts are not well studied in developing countries, particularly in Ethiopia. Therefore, this study aims to assess the spatial distribution of the measles vaccination dropout and its determinants among under-five children in Ethiopia.

### Methods

Data from Ethiopian Demographic and Health Survey 2019 was used for data analysis. The study used a total of 5,753 children. Spatial autocorrelations was used to determine the spatial dependency of measles vaccination dropout. Ordinary interpolation was employed to forecast measles vaccination dropout. Factors associated with measles vaccination dropout were declared significant at p-values <0.05. The data were interpreted using the confidence interval and adjusted odds ratio. A model with the lowest deviance and highest logliklihood ratio was selected as the best-fit model.

### Results

In Ethiopia, one in three under-five children had measles vaccination dropouts. Factors such as birth interval (AOR = 1.87, 95% CI: 1.30, 2.70), unmarried marital status women

**Data Availability Statement:** Anyone may find more information on DHS data and ethical standards by visiting http://www.dhsprogram.com.

**Funding:** The author(s) received no specific funding for this work.

**Competing interests:** The authors have declared that no competing interests exist.

(AOR = 3.98, 95% CI: 1.08, 8.45), $\leq$1 number of under-five children (AOR = 3.86, 95% CI: 2.56, 5.81), rural place of residence (AOR = 2.43, 95% CI: 2.29, 3.11), low community-level ANC utilization (AOR = 3.20, 95% CI: 2.53, 3.56), and residing in Benishangul Gumuz (AOR = 1.80, 95% CI: 1.061, 3.06) had higher odds of measles vaccination dropout.

## Conclusions

Measles vaccination dropout rates in Ethiopia among under-five children were high compared to the maximum tolerable vaccination dropout level of 10% by the WHO. Both individual and community-level variables were determinants of measles vaccination dropout. The ministry of health in Ethiopia should give attention to those mothers of under-five children who reported underutilization of ANC services and rural residences while designing policies and strategies in areas of high spatial clustering of vaccine dropout in Ethiopia.

## Background

Measles is a severe respiratory viral illness that is abrupt and highly contagious. It is characterized by fever, erythematous maculopapular rash, cough, coryza, or conjunctivitis. Four days prior to and four days following the rash's appearance, an infection can spread. The unimmunized population will almost entirely contract the disease [1,2]. Immunization is the process of administering a vaccine to develop immunity against an infectious agent, with the primary goal of preventing the disease caused by the infection [3,4]. Vaccines are essential for the prevention and management of numerous infectious diseases, ensuring the security of world health. Furthermore, they are widely recognized as essential for tackling newly developing infectious diseases, such as by preventing the spread of antibiotic resistance or controlling infectious disease outbreaks [5].

Each year, vaccine-preventable diseases such as diphtheria, tetanus, pertussis, influenza, and measles cost the lives of 8.8 million children under the age of five. In the African region, measles remains a significant cause of disease and death. The estimated number of measles cases increased by 22% from 3,623,869 in 2017 to 4,430,595 in 2021, with a peak of 6,377,451 cases in 2019. Correspondingly, the estimated annual measles deaths increased from 61,166 in 2017 to 66,230 in 2021, with a peak of 104,543 deaths in 2019 [6]. Sub-Saharan Africa and Central and Southern Asia account for about 80% of the world's child deaths because of incomplete vaccination coverage [7–9]. In Ethiopia, measles is endemic, with cases reported annually. Between 12 August 2021 and 1 May 2023, there were 16,814 laboratory-confirmed measles cases and 182 deaths, with a case fatality ratio (CFR) of 1.1%. The annual number of confirmed measles cases has seen a significant increase, from 1,953 in 2021 to 9,291 in 2022, and 6,933 in 2023 as of 1 May. Among the confirmed measles cases, only 36% had received one dose or more of the measles-containing vaccine (MCV). The MCV1 and MCV2 coverage in 2021 was estimated at 54% and 46%, respectively [10].

Childhood immunization, which prevents 1–2 million childhood deaths annually worldwide, is one of the most successful health interventions available to reduce the mortality and morbidity of infectious illnesses in children [11]. The measles vaccine alone prevented 23 million deaths between 2000 and 2018. More than 116 million infants, or 86% of all babies born, receive vaccinations each year, reaching an all-time high [12,13]. Despite this global success, specific challenges persist in regions like sub-Saharan Africa, where 4.4 million children are at

risk due to communicable diseases each year. These risks are exacerbated by inadequate vaccination setups, incomplete vaccination coverage, and barriers to vaccine delivery, which could be mitigated with more robust immunization programs [14,15]. Although all children are expected to receive the measles vaccination at nine months, in Ethiopia, as part of the government's subsidized programme, 3.4% of children did not receive complete measles virus immunization through routine immunization services in 2015 [16]. Ethiopia has also developed a plan to eradicate the measles, with the goal of achieving it by 2020. However, according to the 2019 mini-Ethiopian Demographic Health Survey (EDHS) report, the country's measles vaccination coverage is 59% [17,18].

The interval between the first and second doses of the measles vaccine, known as MCV1 and MCV2, respectively, is a critical factor in ensuring complete immunization. The World Health Organization (WHO) and UNICEF recommend that MCV2 follow MCV1 through routine service strategies. Despite this, MCV2 uptake lags notably behind goals, especially in Sub-Saharan Africa, due to challenges such as healthcare access, maternal education, and poverty [19–21]. Moreover, a significant indicator of an immunization program's success is the measles vaccination dropout rate, which reflects the number of children missing the crucial second dose [22]. High dropout rates, exceeding 10%, signal systemic issues that must be addressed to achieve the Immunization Agenda 2030's objective of universal vaccination coverage [17,23–25].

Despite achievements observed in the reduction of under-five mortality rates, about millions children are still dying each year due to vaccine-preventable diseases, and sub-Saharan African countries share the huge burden of global under-five mortality. As far as our search of the literature and knowledge goes, the risk (hotspot) areas for measles vaccination dropout among under-five children have not been identified in Ethiopia using Ethiopian demographic and health survey.

Therefore, the present study focuses on investigating the spatial distribution and determinants of measles vaccination dropouts among under-five children in Ethiopia using the Ethiopian 2019 Demographic and Health Survey. Identifying geographical variations in measles vaccination dropout is very important to prioritize and design targeted prevention and intervention programmes to prevent measles vaccination dropout at the country level.

## Methods

### Study setting

Ethiopia lies in northeastern Africa, with latitudes ranging from approximately 9.145˚ N to 15˚ N and longitudes from approximately 33˚ E to 48˚ E. Situated on the Horn of Africa, Ethiopia is the continent's second-most populous country. The country is divided into two city administrations, Addis Ababa and Dire Dawa, and nine regional states, namely Amhara, Oromia, Tigray, Benishangul-Gumuz, Somali, Afar, Harari, Southern Nations Nationalities and Peoples (SNNP), and Gambela [26]. A nationally representative sample was used for the survey, which yielded estimates for both rural and urban areas at the national and regional levels. The Ethiopia demographic and health survey (EDHS) is a national and subnational representative household survey, which is conducted every five year. The primary health care unit (PHCU), general hospital, and specialized hospital comprise Ethiopia's three-tiered health system, which prioritizes preventive healthcare. At primary health care units, routine childhood vaccinations are mostly provided through outreach and static destinations [27].

## Study design and period

A secondary data analysis of community-based cross-sectional study was conducted. To get a representative sample from each region of Ethiopia, a recent mini-EDHS 2019 survey was taken to identify factors and the extent of spatial patterns of measles vaccination dropout in Ethiopia. The Ethiopian Demographic and Health Survey (EDHS) is a national-level study conducted every five years using structured, pretested, and validated tools as part of the world-wide Demographic and Health Survey.

## Population and eligibility criteria

Under-five children who are 0–59 months old in Ethiopia were the source population. The study population was all the under-five children who were in the selected enumeration areas included in the analysis.

## Data source and sampling procedure

The Ethiopian Demographic and Health Survey (EDHS 2019) data was used to investigate the spatial distribution of measles vaccination dropouts and determinants among under-five children. The survey contains different datasets, including data on basic health indicators like mortality, morbidity, family planning service utilization, fertility, and maternal and child health services such as vaccination. Using a stratified two-stage cluster design, the Demographic and Health Survey first creates the enumeration regions and then creates a sample of households from each enumeration area in the second stage. For this study, we used the Kids record dataset (KR file) to extract the dependent and independent variables. The variables "received measles-1 (h9) and measles-2 (h9a)" from the kids record (KR) data set were recoded to create the outcome variable (measles vaccination dropout).

## Multi-level analysis

STATA version 14 statistical software was used to extract, clean, record, and analyze the data from the most current EDHS 2019 survey. Before conducting any statistical analysis, the data were weighted using the primary sampling unit, sample weight, and stratum in order to restore the survey's representativeness and account for the sampling design when computing standard errors to produce accurate statistical estimations. We used the weighting variable (v005) as a relative weight normalized to make the analysis survey-specific, while for the pooled data, we denormalized the under-five children's individual standard weight variable by dividing the under-five children's individual standard weight by the sampling fraction: (under-five children adjusted weight = V005) × (total under-five children aged 0–59 years in the country at the time of the survey)/ (number of under-five children aged 0–59 years in the survey).

A binary logistic regression model was used to determine the factors associated with measles vaccination dropout rates. Determinants of the measles vaccination dropout were reported in terms of an adjusted odds ratio (AOR) with a significance level of 95%. In the univariable analysis, at 95% confidence intervals, a p-value of $< 0.25$ was considered a candidate for the multivariable analysis of the data. All variables with p values $<0.05$ were considered statistically significant in multivariable logistic regression.

The assumptions of the standard logistic regression model, such as independence of observations and equal variance, are broken due to the hierarchical nature of the DHS data. Mothers and children, for example, are nested within a cluster, and we assume that the individuals in one cluster may have similar characteristics with those in another, which goes against the equal variance and independence observations between clusters assumptions of the ordinal

logistic regression model. It also suggests that accounting for between-cluster effects requires the use of a complex model. Given this, the parameters associated with measles vaccine dropout were identified using multilevel mixed-effects logistic regression. Multilevel mixed effect logistic regression uses four models: the null model (outcome variable only), model I (only individual level variables), model II (only community level variables), and model III (both individual and community level variables). The null model, which lacks independent variables, was employed to examine the variation in measles vaccine dropout rates within the cluster. Tests were conducted on the relationships between the outcome variable (Model I) and the factors at the individual and community levels (Model II). The association between the community- and individual-level factors and the outcome variable (measles vaccine dropout rate) was fitted simultaneously in the final model, or Model III.

## Spatial analysis

The statistical programmes ArcGIS version 10.8 and Sat-scan version 9.6 have been used for the spatial analysis in order to explore the geographical distribution, spatial autocorrelation, spatial interpolation, and identify significant hotspot areas of measles vaccination dropout. To determine whether or not the spatial distribution of measles vaccination dropout is random, spatial autocorrelation was used. The correlation coefficient for the relationship between a variable and the values around it constitutes the spatial autocorrelation, also known as Global Moran's index. Using the entire data set as input, Mann-index spatial statistics generates a single output that is used to measure spatial autocorrelation. The range of a Moran's index value is -1 to 1. Strong positive spatial autocorrelation is shown by quantities approaching 1, whereas strong negative spatial autocorrelation is indicated by values near -1. In the occurrence that Moran's index is near zero, spatial autocorrelation is absent. If the Moran's index value is statistically significant ($p < 0.05$), the null hypothesis may be rejected (measles vaccination dropout is random), which indicates the presence of spatial dependence.

Using Getis-OrdGi* statistics, significant "hot-spot" and "colds-pot" areas for measles vaccination dropout have been identified. The spatial autocorrelation that varies over the study's location was measured using Getis-OrdGi* statistics and the statistical significance of clustering were determined using the Z-score and p-value estimates. High GI* statistical output denotes a "hot-spot," whereas low GI* evidence denotes a "cold-spot. The importance of using Getis-Ord Gi* statistics over local Moran's I (LISA) lies in its ability to identify specific spatial clusters of high or low values, known as hot-spots and cold-spots. This method is particularly useful when precise geographic targeting is required for interventions. Getis-Ord Gi* is sensitive to small-scale variations and provides a clear distinction between different types of clusters, which can be crucial for policy-making and resource allocation. In contrast, local Moran's I is more suited for detecting general patterns of spatial association without necessarily pinpointing the exact locations of clusters. Therefore, Getis-Ord Gi* is preferred when the research objective is to identify and act upon localized areas of interest within a geographic space [28–30]. Based on the observed values, Kriging interpolation techniques were used to predict the prevalence of measles vaccination dropout among children. While there are numerous types of interpolation techniques, we decided to use conventional Kriging methods for this study because of its low residuals and root mean square error.

Sat scan version 9.6 Software was used to do a spatial Sat scan analysis in order to identify significant primary and secondary clusters. Since the result variable was binary, Bernoulli's model was fitted. A child who did not have a measles vaccination dropout was categorized as a control, while those who had a vaccination dropout were classified as cases. Data for cases, controls, and geographic locations are required for the Bernoulli model. Clusters containing

more than the maximum limit were ignored. Both small and significant clusters may be identified using the default maximum spatial cluster size of <50% of the population as an upper limit. The most likely cluster was identified as the scanning window with maximum likelihood, and each cluster was given a p-value by the application of Monte Carlo hypothesis testing [31].

## Study variables

**Dependent variables.** The outcome of this study was measles vaccination dropout, in which the child who received the first dose of the measles but not the second dose of the measles vaccination [32]. The measles vaccination dropout was assessed by recoding the variables received measles-1 (h9) and received measles-2 (h9a) from the Kids Record (KR) data set. The measles vaccination dropout rate was calculated by dividing the number of children aged 0–59 months who received measles 1 by the number of children aged 0–59 months who received measles 2 divided by the number of children 0–59 months of age who received measles 1 multiplied by 100%; $= \frac{Number\ of\ childreh\ who\ recieved\ measles\ vaccine\ 1\ - Number\ of\ children\ who\ recieved\ measles\ vaccine\ 2}{Number\ of\ childreh\ who\ recieved\ measles\ vaccine\ 1} \times 100\%$ [33].

**Independent variables.** Independent variables from two sources (variables at the individual and community levels) were taken into account for this analysis because DHS data are hierarchical. The individual-level independent variables were: Sex of child (Male, Female), Sex of household head (Male, Female), Birth interval (months) (<24, 24–48, >48), Maternal age (15–24, 25–34, 35–49), Maternal educational status (No formal education, Primary, Secondary and higher), Marital status of the mother (Unmarried, Married, Ever married), Place of delivery (Home, Health facility), Mode of delivery (vaginal delivery, caesarean section delivery), Number of ANC visits (<4, ≥4), Postnatal checkup within 2 months (No, Yes), Number of under-5 children (≤1, 2, ≥3), Household wealth index (Poor, Middle, Rich). The community-level variables were Place of residence (Urban, Rural), Community level women illiteracy (Low, High), Community level poverty (Low, High), Community level ANC utilization (Low, High), and region (Amhara, Oromia, Tigray, Benishangul-Gumuz, Somali, Afar, Harari, Southern Nations Nationalities and Peoples (SNNP), and Gambela) (Fig 1).

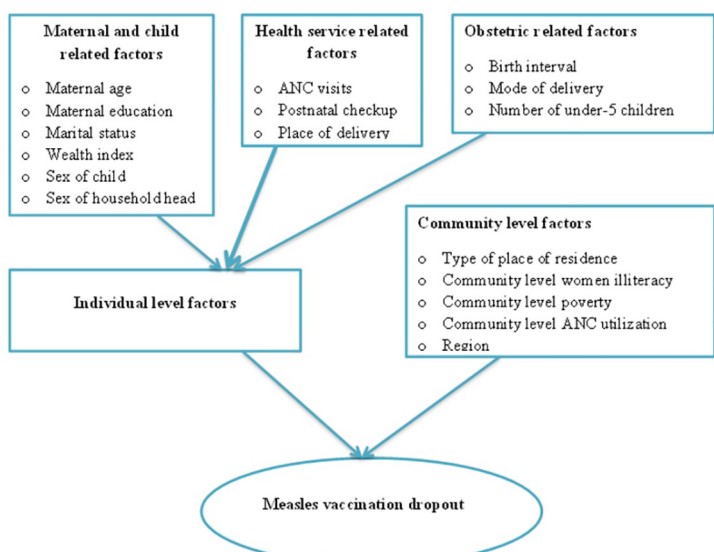

**Fig 1. Conceptual framework for factors associated with measles vaccination dropouts among under-five children in Ethiopia.**

## Operationalization of independent variables

**Wealth index.**   The wealth index is a composite indicator of a household's cumulative living level. Easy-to-collect information on a household's possession of certain goods, such televisions and bicycles, building materials used to construct homes and kinds of water access and sanitary facilities are utilized to create the wealth index [34].

**Community level women illiteracy.**   The proportion of women having a minimum primary level of education is generated from data on respondents' levels of education. Then, it was categorized using the national mean value after computation of cross tabulating the individual level of women's education with the cluster number (v001): low community-level women's illiteracy (communities with $\geq$50% of the national mean value of women's education) and high community-level women's illiteracy (communities with $\geq$50% of the national mean value of women's community illiteracy).

**Community level poverty.**   It is generated from proportion of women in both rich and middle-wealth categories. After computing the cross-tabulating individual-level combined wealth index with the cluster number (v001), it was then classified using the national mean value of the wealth index: low community-level poverty (communities with $\geq$50% of the national mean value of the community wealth index) and high community-level poverty (communities with <50% of the national mean value of the community wealth index).

## Random effects (Measures of variation)

Random effects or measures of variation such as Likelihood Ratio test (LR), Intra-class Correlation Coefficient (ICC), and Median Odds Ratio (MOR) were computed to measure the variation of measles vaccination dropout rates across clusters. Taking clusters as a random variable, the ICC quantifies the degree of heterogeneity of measles vaccination dropout rates between clusters (the proportion of the total observed variation in pentavalent dropout that is attributable to between cluster variations) [35] is computed as; $ICC = \frac{VC}{VC+3.29} \times 100\%$. The Median Odds Ratio (MOR) is the median value of the odds ratio which quantifies the variation or heterogeneity in measles vaccination dropout rates between clusters in terms of odds ratio scale and is defined as the median value of the odds ratio between the cluster at high likelihood of measles vaccination dropout rates and cluster at lower risk when randomly picking out individuals from two clusters [36]; $MOR = e^{0.95\sqrt{VC}}$.

Moreover, the proportional change in variance (PCV) demonstrates the variation in the measles vaccination dropout rates explained by determinants and computed as; $PCV = \frac{Vnull-Vc}{Vnull} \times 100\%$; where Vnull = variance of the null model and VC = cluster level variance [37]. The fixed effects were used to estimate the association between the likelihood of measles vaccination dropout rates and individual and community level independent variables. It was assessed and the strength was presented using adjusted odds ratio (AOR) and 95% confidence intervals with a p-value of < 0.05. Deviance = -2 (log likelihood ratio) was used to compare the models due to the nested nature of the model; the model with the lowest deviance and the highest log likelihood ratio was chosen as the best-fit model. By calculating the variance inflation factors (VIF), the multi-collinearity of the variables employed in the models was confirmed, and the results were found to be within reasonable bounds of one to ten.

**Ethical approval and consent to participate.**   Since this study is merely a secondary review of the DHS data, ethical approval is not needed. We enrolled with the DHS web archive, requested the dataset for our study, and were granted permission to view and download the data files. As per the DHS study, all participant data were anonymized at the time of survey data collection. Visit https://www.dhsprogram.com for additional information on DHS data and ethical standards.

## Result

### Socio-demographic and economic characteristics of under-five children in Ethiopia

A total of 5,753 under-five children (2,969 males and 2,784 females) were included in the analysis. More than half 3,149 (54.74%) of the under-five children were born to mothers with no formal education. More than three-fourth 4,425 (76.92%) of the under-five children was born to mothers living in rural areas of Ethiopia. Nearly half of (48.76%) of the children were born to mothers living in a high community poverty level (Table 1).

### Prevalence of measles vaccination dropout among under-five children in Ethiopia

The overall measles vaccination dropout rate in Ethiopia was 33.3% (95% CI: 31.6, 34.9). The urban and rural measles vaccination dropout rates in Ethiopia were found to be 34% and 66%, respectively (Fig 2). The measles vaccination dropouts among under-five children were significantly varied across the regions in Ethiopia. Subsequently, the lowest measles vaccination dropout was observed in Afar region (5.1%), while the highest was seen in the Benishangul Gumuz region which was 12.2% (Fig 3).

### Spatial distribution of measles vaccination dropout among under-five children in Ethiopia

**Spatial autocorrelation of measles vaccination dropout.** There was significant variation in the measles vaccine dropout rate across the regions (Moran's index = 0.358552, p-value <0.001). According to the spatial autocorrelation information, there was a clustering effect in measles vaccination dropout, meaning that there were high dropout rates in particular areas and low dropout rates in others. The outputs on the left and right sides of the panel have established keys. The clustered pattern's z-score of 7.859437 indicates that the likelihood of it being a random coincidence is less than 1% (Fig 4).

**Hotspot analysis of measles vaccination dropout.** The local Getis-Ord Gi* statistics were employed in the present study to determine the measles vaccination dropout hot and cold spots. Significant hot spot (high-risk) areas for measles vaccination dropout are indicated by the colors red and orange, while cold spot (low-risk) areas are indicated by the color green. The geographic areas of northwestern Addis Ababa, central and southwest Benishangul Gumuz, southern Amhara, central and northwest Tigray, northern SNNPR, northern Oromia had the highest rates of measles vaccination dropouts. However, the regions with the lowest distribution of measles vaccine dropouts were, Afar, Gambela, Somali, Dire Dawa, and Harari (Fig 5).

**Interpolation of measles vaccination dropout.** In the Kriging interpolation, the predicted high prevalence of measles vaccination dropout was identified in the northwestern, northern, central, and eastern parts of Ethiopia. The southern, northwestern, southwest, and southeast parts of the country were predicted to be areas of low measles vaccination dropout (Fig 6).

**Sat scan statistical analysis of measles vaccination dropout.** In the spatial scan statistics, a total of 166 significant clusters were identified; of these, 113 were primary (most likely) clusters. The primary clusters were located in Benishangul Gumuz, Addis Ababa, Tigray, and Amhara regions, centered at 10.833913 N, 36.813679 E of geographic location with a 357.71km radius, a relative risk (RR) of 1.66, and a log-Likelihood ratio (LLR) of 50.36 at p < 0.001. It showed that children within a spatial window had a 1.66 times higher likelihood

**Table 1. Socio-demographic and economic characteristics of under-five children in Ethiopia.**

| Individual level variables | Category | Frequency (n) | Percent (%) |
|---|---|---|---|
| Sex of child | Male | 2,969 | 51.61 |
| | Female | 2,784 | 48.39 |
| Sex of household head | Male | 4,598 | 79.92 |
| | Female | 1,155 | 20.08 |
| Birth interval (months) | <24 | 1,091 | 24.39 |
| | 24–48 | 2,224 | 49.72 |
| | >48 | 1,158 | 25.89 |
| Maternal age | 15–24 | 1,439 | 25.01 |
| | 25–34 | 3,097 | 53.83 |
| | 35–49 | 1,217 | 21.15 |
| Maternal educational status | No formal education | 3,149 | 54.74 |
| | Primary | 1,823 | 31.69 |
| | Secondary and higher | 781 | 13.58 |
| Marital status of the mother | Unmarried | 31 | 0.54 |
| | Married | 5,396 | 93.79 |
| | Ever married | 326 | 5.67 |
| Place of delivery | Home | 2,955 | 51.36 |
| | Health facility | 2,798 | 48.64 |
| Mode of delivery | vaginal delivery | 5,404 | 93.93 |
| | caesarean section delivery | 349 | 6.07 |
| Postnatal check within 2 months | Yes | 532 | 13.37 |
| | No | 86.63 | 86.63 |
| Number of ANC visits | No visit | 1,061 | 26.66 |
| | <4 | 1,262 | 31.72 |
| | ≥4 | 1,656 | 41.62 |
| Number of under-5 children | ≤1 | 2,297 | 39.93 |
| | 2 | 2,504 | 43.53 |
| | ≥3 | 952 | 16.55 |
| Household wealth index | Poor | 2,958 | 51.42 |
| | Middle | 805 | 13.99 |
| | Rich | 1,990 | 34.59 |
| **Community level variables** | | | |
| Place of residence | Urban | 1,328 | 23.08 |
| | Rural | 4,425 | 76.92 |
| Community level women illiteracy | Low | 3,077 | 53.49 |
| | High | 2,676 | 46.51 |
| Community level poverty | High | 2,805 | 48.76 |
| | Low | 2,948 | 51.24 |
| Community level ANC utilization | Low | 2,380 | 41.37 |
| | High | 3,373 | 58.63 |

(*Continued*)

**Table 1.** (Continued)

| Individual level variables | Category | Frequency (n) | Percent (%) |
|---|---|---|---|
| Region | Tigray | 454 | 7.89 |
| | Afar | 652 | 11.33 |
| | Amhara | 511 | 8.88 |
| | Oromia | 719 | 12.5 |
| | Somali | 637 | 11.07 |
| | Benishangul | 530 | 9.21 |
| | SNNPR | 660 | 11.47 |
| | Gambela | 450 | 7.82 |
| | Harari | 447 | 7.77 |
| | Addis Ababa | 291 | 5.06 |
| | Dire Dawa | 402 | 6.99 |

of high measles vaccination dropout than under-five children outside the spatial window (Fig 7) (Table 2).

## Random effect ((Measures of variation) and model fitness

With a variance of 0.8811035, the null model's findings revealed that there were significant variations in measles vaccine dropout rates across communities. According to the null model, community-level factors account for approximately 21.1% of the overall variation in measles vaccine dropout that occurs at the cluster level. Furthermore, when an individual is randomly selected from one cluster at a higher risk of measles vaccination dropout and another cluster at a lower risk, the null model has the highest median odds ratio (MOR) value (2.44). This means that individuals in the higher risk cluster have 2.44 times higher odds of being measles vaccination dropouts than their counterparts. According to Model I's intraclass correlation value, differences between communities are explained by 10.1% of the variation in measles vaccine dropout rates. Next, using the null model, we created Model II using variables at the community level. Based on the ICC value from Model II, cluster variations accounted 7.47% of the variation in the measles vaccine dropout rate. The final model, known as model III, attributed both individual and community-level factors approximately 77.74% of the variation in the likelihood of measles vaccine dropout rate. The final model, model III was the best-fitted model

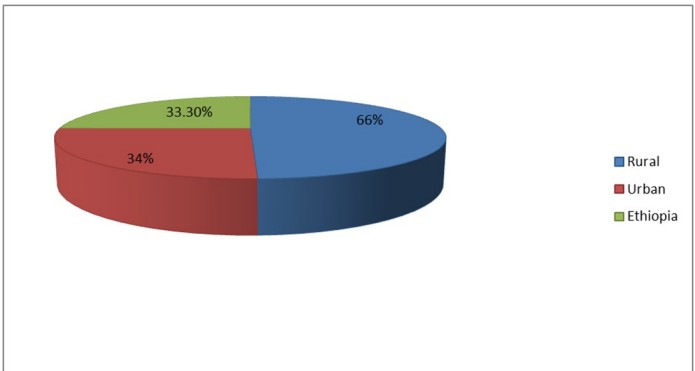

**Fig 2. Measles vaccination dropout rates among under-five children in Ethiopia, Mini EDHS 2019.**

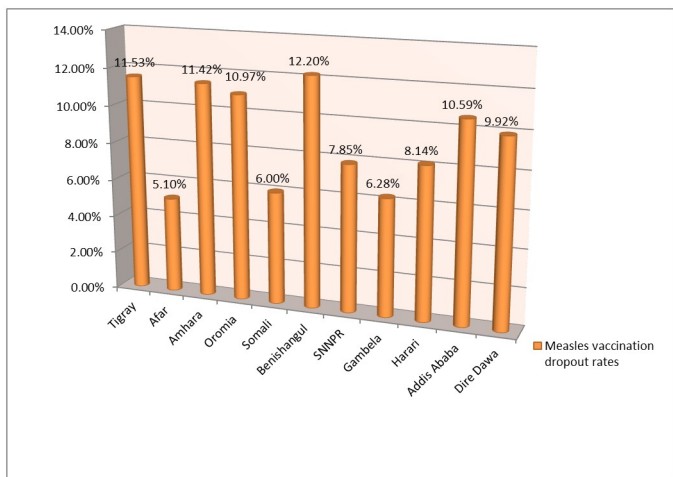

**Fig 3. Regional prevalence of measles vaccination dropout rates among under-five children in Ethiopia, Mini EDHS 2019.**

since it had the lowest deviance (2497.216) and the highest logliklihood ratio (-1248.608). Then the model fitness has been determined using these parameters (Table 3).

## Association of individual and community-level determinants and measles vaccination dropouts among under-five children in Ethiopia

In multivariable multilevel logistic regression analysis, where both the individual and community level factors were fitted simultaneously, birth interval, marital status of the mother,

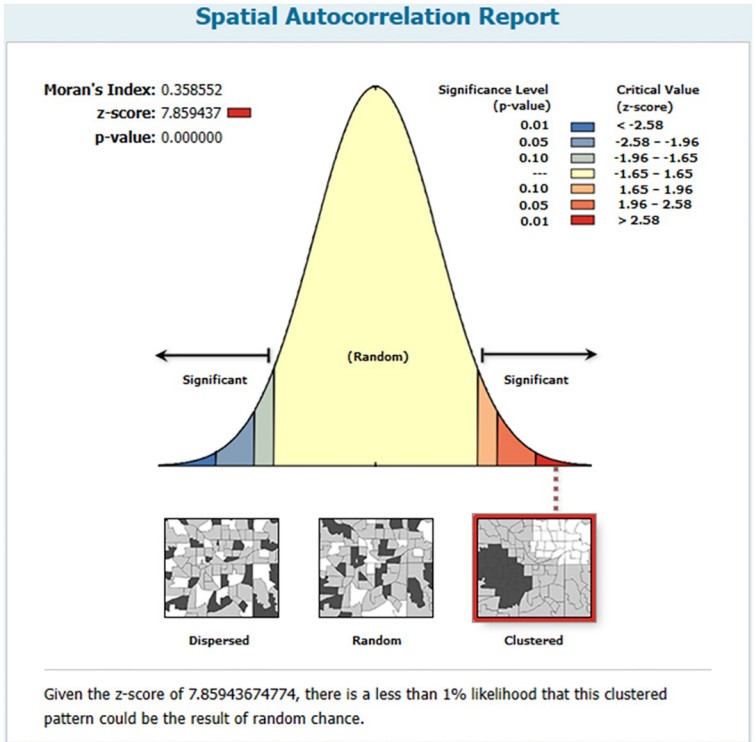

**Fig 4. Spatial autocorrelation of measles vaccination dropout in Ethiopia based on feature locations and attribute values using the Global Moran's index statistic, Mini EDHS 2019.**

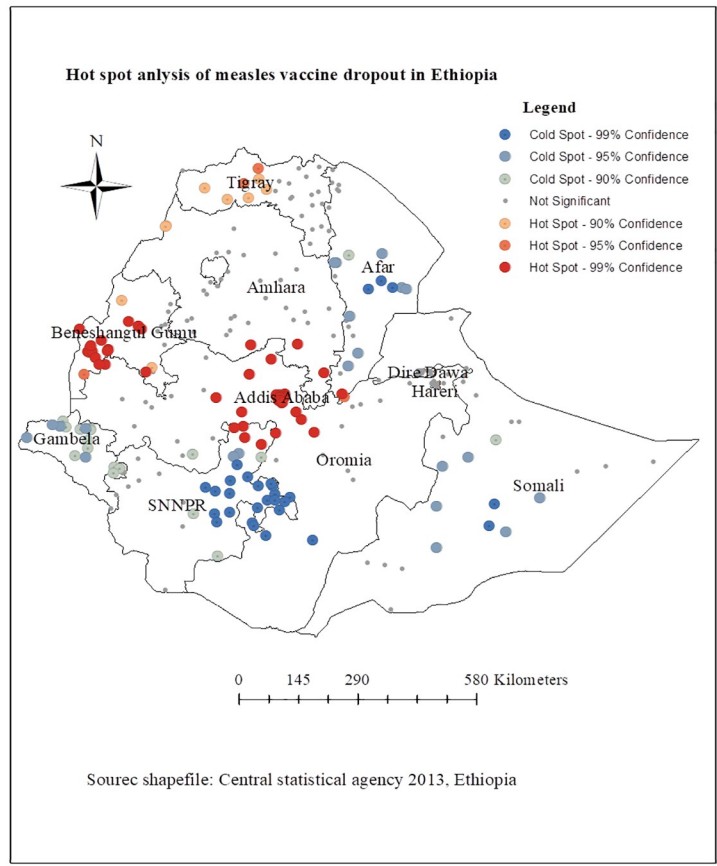

**Fig 5. Hotspot analysis of measles vaccination dropout in Ethiopia, Mini EDHS 2019.**

number of under-five children, place of residence, community level ANC utilization, and region (Benishangul Gumuz) were significantly associated with measles vaccination dropout at 95% confidence interval and a p-value of <0.05.

The odds of measles vaccination dropout rates were 1.87 times higher among under-five children whose birth interval was less than 24 months compared with under-five children whose birth interval was greater than 48 months (AOR = 1.87, 95% CI: 1.30, 2.70). Measles vaccination dropout rates were 3.98 times higher among children under five born to unmarried mothers than among children under five born to married mothers (AOR = 3.98, 95% CI: 1.08, 8.45). The odds of measles vaccination dropout rates were 3.86 times higher among under-five children, where the number of under-five children is one and less compared with the number of under-five children of three or more (AOR = 3.86, 95% CI: 2.56, 5.81). Measles vaccination dropout was 2.43 times higher among under-five children whose place of residence was rural as compared to under-five children from urban residence (AOR = 2.43, 95% CI: 2.29, 3.11).

The odds of measles vaccination dropout rates were 3.20 times higher among under-five children who had high community-level ANC utilization as compared to under-five children whose community-level ANC utilization was low (AOR = 3.20, 95% CI: 2.53, 3.56). Under-five children whose residence is in the Benishangul Gumuz region have a 1.80 times higher measles vaccination dropout rate compared to the Amhara region (AOR = 1.80, 95% CI: 1.061, 3.06) (Table 3).

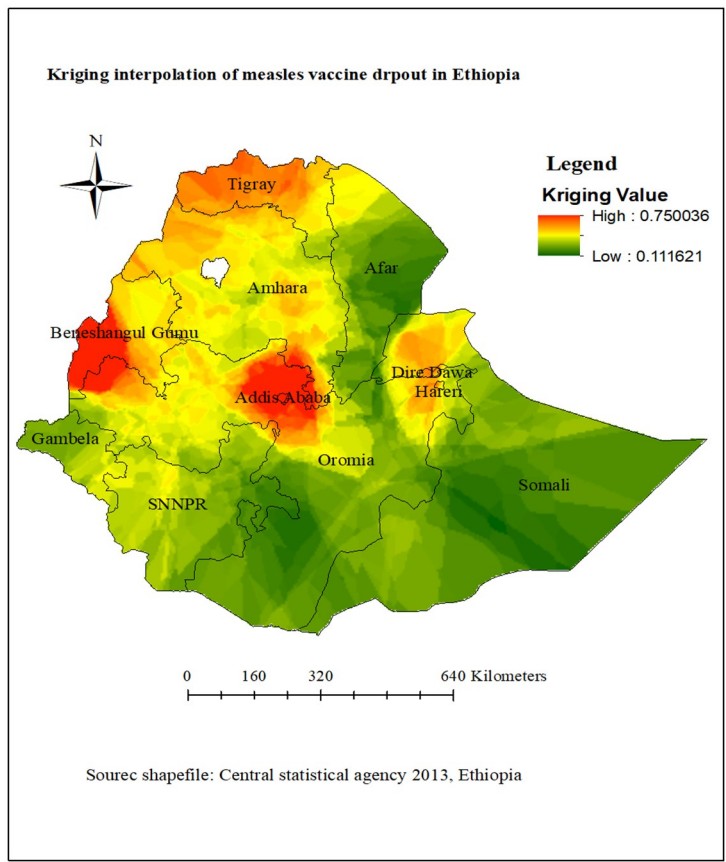

**Fig 6. Kriging interpolation analysis of measles vaccination dropout in Ethiopia, Mini EDHS 2019.**

## Discussion

Measles is preventable with two doses of the measles vaccine. Despite a modest improvement in vaccination rates globally from 2021 to 2022, 33 million children were still not vaccinated against measles: around 22 million missed their first dose, and an additional 11 million, or 74% of the total, missed their second dose [38]. Although there has been considerable progress toward measles elimination, Ethiopia remains among the countries with the highest number of children missing their initial dose of the measles vaccine [39]. Surveys identified over 500 children aged 12–23 months as never-been-vaccinated (zero-dose) and over 1,800 as under-vaccinated in regions of Ethiopia [40]. This study used data from the EDHS 2019 to determine the prevalence, geographic distribution, and determinants of measles vaccine dropout in Ethiopia.

In this study, the prevalence of measles vaccination dropout among under-five children in Ethiopia was found to be 33.26%. This finding was consistent with a previous study conducted in Nigeria, 33.59% [41], and Ghana, 31.5% [42]. This could be due to Ethiopia, Nigeria, and Ghana has similar health policies regarding measles vaccination, aiming for high coverage through national immunization programs. However, challenges such as vaccine supply chain issues and inadequate health funding may contribute to the high dropout rates observed [43]. The prevalence of the measles vaccination dropout rate in this study was higher than the findings conducted in Gambia, 7% [44], South Africa, 4.1% [45], India, 27.7% [46], and 18.6%

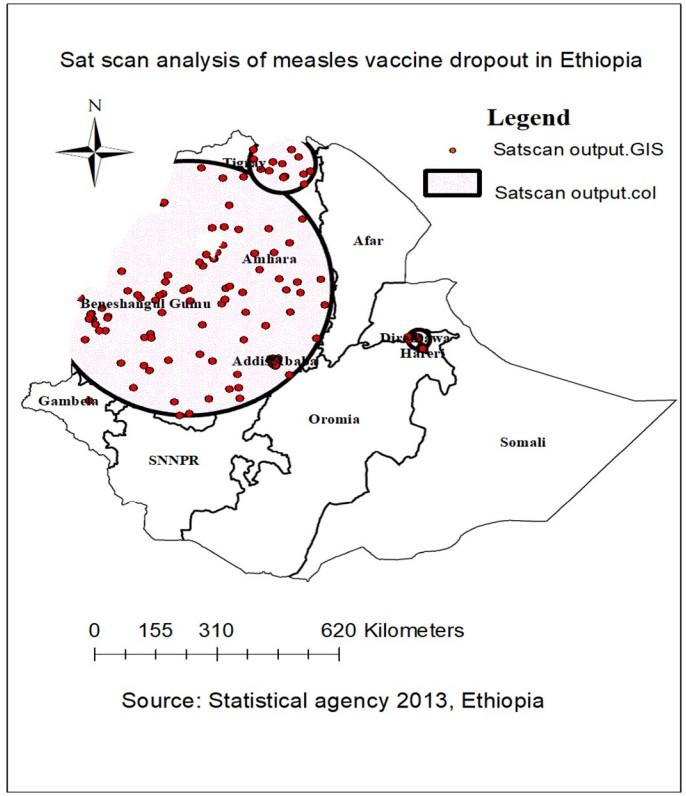

**Fig 7. Sat scan analysis of measles vaccination dropout in Ethiopia, Mini EDHS 2019.**

[47]. On the other hand, the prevalence of measles vaccination dropout rate in this study was lower than the findings conducted in Amritsar city, India, at 38.5% [48]. The potential explanation for these discrepancies may be due to differences in health policy, quality of health care, and socioeconomic and cultural differences across the countries [44,49–51].

Gambia and South Africa have more robust health policies and better implementation strategies, resulting in lower dropout rates. Gambia, for instance, has achieved significant success

**Table 2. Sat scan analysis of measles vaccination dropout among children under-five children in Ethiopia: EDHS 2019.**

| Cluster | Enumeration areas | Coordinates/ radius | Population (n) | Cases (n) | RR | LLR | P-value |
|---|---|---|---|---|---|---|---|
| 1(113) | 79, 80, 77, 162, 165, 163, 53, 148, 75, 166, 52, 76, 164, 72, 70, 81,54, 74, 119, 161, 59, 158, 71, 57, 160, 73, 167, 168, 159, 169, 84, 60, 98, 58, 82, 99, 83, 93, 87, 150, 156, 100, 149, 65, 61, 86, 92,51, 85, 55, 120, 157, 155, 154, 153, 151, 56, 152, 78, 147, 67, 118,146, 112, 63, 66, 62, 262, 259, 257, 258, 256, 261, 260, 265, 263, 266, 267, 276, 264, 170, 275, 273, 269, 270, 268, 277, 274, 271, 174, 272, 279, 171, 278, 280, 94, 95, 97, 22, 64, 176, 21, 9, 68, 18, 69, 101, 91, 96 | (10.833913 N,36.813679 E)/ 357.71km | 986 | 453 | 1.66 | 50.36 | <0.0001 |
| 2(13) | 11, 12, 14, 13, 2, 16, 10, 7, 17, 1, 3, 5, 6 | (13.985245N,38.954807E)/ 81.73 km | 126 | 70 | 1.72 | 13.72 | <0.0001 |
| 3(40) | 300, 299, 301, 298, 251, 295, 293, 290, 289, 253, 302, 294, 292, 297, 238, 232, 291, 236, 288, 285, 286, 231, 296, 240, 287, 239, 246, 303, 237, 283, 235, 242, 284, 243, 233, 241, 234, 282, 252, 245 | (9.537839N, 42.066551E)/ 27.52 km | 347 | 156 | 1.41 | 11.51 | <0.0028 |

LLR, Lo-likelihood ratio; RR, Relative risk.

**Table 3. Multivariable multilevel logistic regression analysis of individual-level and community level factors associated measles vaccination dropout rates among under-five children in Ethiopia, Mini EDHS 2019.**

| Individual level factors | Category | Model I AOR(95% CI) | Model II AOR(95% CI) | Model III AOR(95% CI) |
|---|---|---|---|---|
| Sex of child | Male | 1 | | 1 |
| | Female | 0.97(0.79, 1.18) | | 0.95(0.78, 1.16) |
| Sex of household head | Male | 1 | | 1 |
| | Female | 1.32(0.99, 1.75) | | 1.40 (0.05, 1.88) |
| Birth interval (months) | <24 | 1.72(1.27, 2.35) | | **1.87(1.30. 2.70)** |
| | 24–48 | 0.96(0.73, 1.26) | | 0.93(0.70, 1.22) |
| | >48 | 1 | | 1 |
| Maternal age | 15–24 | 0.75(0.53, 1.06) | | 0.84(0.59, 1.20) |
| | 25–34 | 0.98(0.76, 1.27) | | 1.02(0.79, 1.31) |
| | 35–49 | 1 | | 1 |
| Maternal educational status | No formal education | 0.89(0.61, 1.29) | | 1.10(0.85, 1.41) |
| | Primary | 1.06(0.74, 1.51) | | 0.92(0.62, 1.35) |
| | Secondary and higher | 1 | | 1 |
| Marital status of the mother | Unmarried | 4.27(1.32, 9.52) | | **3.98(1.08, 8.45)** |
| | Married | 1 | | 1 |
| | Ever married | 0.92(0.53, 1.59) | | 0.20(0.69, 2.07) |
| Place of delivery | Home | 0.70(0.54, 0.91) | | 0.83(0.64, 1.08) |
| | Health facility | 1 | | 1 |
| Mode of delivery | vaginal delivery | 0.74(0.48, 1.14) | | 0.67(0.43, 1.04) |
| | CS | 1 | | 1 |
| Number of ANC visits | No visit | 0.48(0.35, 0.66) | | 0.56(0.40, 1.78) |
| | <4 | 0.79(0.62, 0.99) | | 0.89(0.70, 1.12) |
| | ≥4 | 1 | | 1 |
| Postnatal checkup within 2 months | No | 0.92(0.69, 1.23) | | 0.92(0.69, 1.23) |
| | Yes | 1 | | 1 |
| Number of under-5 children | ≤1 | 3.98(2.64, 6.02) | | **3.86(2.56, 5.81)** |
| | 2 | 1.13(0.53, 2.98) | | 1.06(0.48, 2.88) |
| | ≥3 | 1 | | 1 |
| Household wealth index | Poor | 0.76(0.58, 1.01) | | 1.09(0.77, 1.54) |
| | Middle | 0.67(0.48, 0.94) | | 0.84(0.58, 1.21) |
| | Rich | 1 | | 1 |
| **Community level variables** | | | | |
| Place of residence | Urban | | 1 | 1 |
| | Rural | | 2.76(2.56, 3.72) | **2.43(2.29, 3.11)** |
| Community level women illiteracy | Low | | 1 | 1 |
| | High | | 1.26(0.98, 1.63) | 1.28(0.95, 1.72) |
| Community level poverty | High | | 1.34(1.03, 1.74) | 1.19(0.86, 1.65) |
| | Low | | 1 | 1 |
| Community level ANC utilization | High | | 1 | 1 |
| | Low | | 3.61(3.46, 3.80) | **3.20(2.53, 3.56)** |

(*Continued*)

**Table 3.** (Continued)

| Individual level factors | Category | Model I AOR(95% CI) | Model II AOR(95% CI) | Model III AOR(95% CI) |
|---|---|---|---|---|
| Region | Tigray | | 1.06(0.65, 1.74) | 0.73(0.39, 1.38) |
| | Afar | | 0.29(0.17, 0.49) | 0.31(0.17, 0.56) |
| | Amhara | | 1 | 1 |
| | Oromia | | 0.62(0.39, 0.99) | 0.60(0.35, 1.04) |
| | Somali | | 0.61(0.36, 1.06) | 0.74(0.39, 1.37) |
| | Benishangul | | 1.88(1.77, 1.90) | **1.80 (1.061, 3.06)** |
| | SNNPR | | 1.44(0.27, 0.72) | 1.51(0.28, 0.93) |
| | Gambela | | 1.48(0.29, 0.80) | 1.47(0.24, 0.91) |
| | Harari | | 1.60(0.36, 0.98) | 1.51(0.28, 0.90) |
| | Addis Ababa | | 1.39(0.82, 2.37) | 1.13(0.59, 2.17) |
| | Dire Dawa | | 0.97(0.61, 1.56) | 1.52(0.83, 2.78) |
| Random effects | | | | |
| Parameter | Null model | Model I | Model II | Model III |
| Variance | 0.8811035 | 0.3683643 | 0.2656147 | 0.1961003 |
| ICC | 21.1% | 10.1% | 7.47% | 5.63% |
| MOR | 2.44 | 1.78 | 1.63 | 1.52 |
| PCV | Reference | 58.19% | 69.85% | 77.74% |
| **Model fitness** | | | | |
| LLR | -1946.0799 | -1276.3637 | -1861.8774 | -1248.608 |
| Deviance | 3892.1598 | 2552.7274 | 3723.7548 | 2497.216 |

ICC: Interacluster correlation, LLR: Logliklihood ratio, MOR: Median odds ratio, PCV: Proportional change in variance.

towards measles elimination, indicating effective policy execution [52]. India has a vast and diverse healthcare system with varying policies across states, which might explain the differences in dropout rates between the national average and specific cities like Amritsar [43]. The quality of healthcare in Sub-Saharan Africa, including Ethiopia, is often compromised by limited healthcare infrastructure and workforce, affecting the continuity and completion of measles vaccination schedules [19].

In spatial analysis, the measles vaccine dropout rate was not randomly distributed in the regions of Ethiopia. This is consistent with previous study conducted in Butajira, Ethiopia [53]. The spatial autocorrelation statistic confirmed that the distribution of measles vaccination dropout was clustered in some geographical areas of Ethiopia. According to the spatial hot spot analysis, geographic variations in measles vaccine dropout have been identified in the northwest, northern, central, and eastern regions of Ethiopia. The most plausible explanation for this geographical variance in measles vaccine dropout rates could be due to significant differences in infrastructure, such as the distribution of healthcare facilities and professionals among regions, as well as differences in roads, electricity, and water [54,55]. In addition, there are regional variations in the distribution of vaccines, culture, maternal Socio-demographic factors, and societal attitudes and levels of awareness regarding immunization [56,57].

Three likely significant clusters of areas with high measles vaccination dropout rates were identified using the Sat Scan analysis across the study area. This suggests that children living in those geographic clusters of areas had a higher chance of measles vaccination dropout than children located outside the spatial scan window. The results of the multivariable multilevel

analysis of EDHS 2019 identified the significant factors that are associated with measles vaccination dropout in Ethiopia. In multilevel logistic regression analysis, birth interval, marital status of the mother, number of under-five children, place of residence, community-level ANC utilization, and region (Benishangul Gumuz) were significantly associated with measles vaccination dropout.

The odds of measles vaccination dropout rates were 1.87 times higher among under-five children whose birth interval was less than 24 months compared to under-five children whose birth interval was greater than 48 months. This finding is supported by the previous study conducted in Kenya [58]. The possible explanation might be that short birth intervals can cause a burden on a woman's ability to care for her dependent children as well as other responsibilities in the home. Because of this, mothers might not have the time or desire to bring their children to the health institution to get essential vaccines [59,60]. As a result, children under the age of five who do not receive vaccinations or do not finish their booster series become vaccine dropouts [61,62].

Measles vaccination dropout rates were 3.98 times higher among under-five children born from unmarried mothers as compared to under-five children delivered from mothers who had married. It is in line with study findings in Ethiopia [63,64] and Ghana [65]. This may be due to the fact that married women can get husband support in making decisions about child vaccination. On the other hand, compared with unmarried mothers, married women have greater social connections and are more likely to communicate with their spouses and peers about child vaccinations, which may help mothers remember the schedule [66,67].

The odds of measles vaccination dropout rates were 3.86 times higher among under-five children where the number under-five children is one and less compared to the number under-five children three or more. This is supported by the previous studies conducted in Zimbabwe [68]. The possible explanation could be that a large family size is likely to strengthen maternal capabilities to extend more care to the younger children as well as her mobility to get access to immunization services due to experience [69].

On the contrary, studies from Ghana [70], Australia [71], India [72], Lebanon [73], and Brazil [74] revealed a weak association between a small number of under-five children and vaccination dropouts. Due to the fact that large number of under-five children is likely to hinder maternal capabilities to extend more care to the younger children as well as her mobility to get access to immunization services. The discrepancies could be variations in national immunization programs, healthcare infrastructure, cultural practices, and socioeconomic status, which can significantly influence vaccination rates [75].

Measles vaccination dropout was 2.43 times higher among under-five children whose place of residence was rural as compared to under-five children from urban residence. This is consistent with the findings from previous studies in Malawi [76]. The possible explanation might be that the low vaccination dropout rate among urban children may be due to the following factors: highly educated mothers may have greater control and autonomy over residence resources; they may also have changed traditional perspectives about immunization. As a result, they might be more likely to seek healthcare and acquire new information about health issues earlier [77–79]. Moreover, child vaccination is influenced by the wealth index of the household, mothers' antenatal care visits, region, and possession of a vaccination card of rural residence women. Additionally, mothers from wealthier homes might find it easier to obtain immunization services at healthcare facilities than families from lower-income backgrounds [55,80].

The findings of this study are contradictory to those of the studies conducted in Ethiopia [41,81], Nigeria [82], Uganda [83], India [84] and Bangladesh [85]. The possible explanation for the inconsistency might be low vaccination dropouts in rural areas are due to the use of

traditional birth attendants and primary health care workers, who both play a role in encouraging mothers to attend the maternal and child health clinics, although these roles do not formally exist in urban areas. Another possible reason why vaccination coverage is high and dropouts are low in rural areas might be due to the establishment and use of outreach clinics. It is established that sustained outreach is an approach for reaching remote areas of the population with limited access to vaccination centers [86,87].

The odds of measles vaccination dropout rates were 3.20 times higher among under-five children who have low community level ANC utilization as compared to under-five children whose community level ANC utilization was high. This finding is in line with other studies in Ethiopia [88] and Gambia [44]. This association might be a result of the fact that mothers who do not make full use of ANC services miss out on opportunities to get information about the importance and timing of child vaccinations. In addition, mothers who decide not to receive antenatal care or give birth in a healthcare facility might not give the same attention to childhood vaccinations as their peers from similar socioeconomic backgrounds [89–91].

Under-five children whose residence is in the Benishangul Gumuz region have a 1.80 times higher measles vaccination dropout rate compared to the Amhara region. The possible explanation for the association might be that even though vaccinations are free in Ethiopia, mothers must still cover certain indirect costs (such as food and transportation expenses), which frequently restrict their use of maternity and child health services, including immunization. In this sense, mothers in the lower income category are more likely to encounter these obstacles when trying to get to healthcare services, which eventually results in immunization dropouts [92,93]. Our finding is consistent with studies conducted in Nigeria [94] and Benin [95], where existing vaccination dropout disparities disproportionately disadvantage the poor.

## Strength and limitations of the study

The study's strength was the utilization of a recently conducted large-sample national demography and health survey to show the spatial distribution of measles vaccination dropout in Ethiopia. The EDHS is a comprehensive source of data that is representative at both national and regional levels. Its large sample size enhances the statistical power of our analysis, allowing for a robust spatial distribution mapping of measles vaccination dropout rates across Ethiopia. The study's utilization of mixed multilevel logistic regression to identify two-level factors that was not possible with standard logistic regression was another strength. However, the study was limited in its ability to include other variables that might have been associated with the outcome variable due to a lack of some important variables in the Ethiopian mini-EDHS 2019, such as media exposure, distance to a health facility, and mother's psychological factors of the mother. While these variables are indeed pertinent, the mini-EDHS 2019 did not collect data on them. To mitigate this limitation, we have discussed potential proxies for these variables that were available in the dataset, such as education level and wealth index, which may serve as indirect indicators of media exposure and access to healthcare services.

## Conclusions and recommendation

This study concludes that measles vaccination dropout rates in Ethiopia among under-five children were high compared to the maximum tolerable vaccination dropout level of 10% by the WHO. The study identified that both individual and community-level variables were determinants of measles vaccination dropout rates. Therefore, the Government and ministry of health in Ethiopia should give attention to those mothers of under-five children who reported underutilization of antenatal care services and rural residences while designing policies and strategies targeting reducing measles vaccination dropout rates in areas of spatial

clustering in Ethiopia. Targeted Interventions: Focusing on mothers who have reported under-utilization of antenatal care services, particularly in rural areas. Policy Revisions: Implementing policy changes that prioritize resource allocation to regions with high spatial clustering of vaccination dropouts. Community Engagement: Encouraging community-level health education programs to raise awareness about the importance of complete vaccination schedules.

## Supporting information

**S1 File.**
(RAR)

**S2 File.**
(RAR)

**S3 File.**
(RAR)

**S1 Data set.**
(RAR)

## Author Contributions

**Conceptualization:** Alebachew Ferede Zegeye, Berhan Tekeba, Almaz Tefera Gonete, Tadesse Tarik Tamir.

**Data curation:** Alebachew Ferede Zegeye, Enyew Getaneh Mekonen, Belayneh Shetie Workneh.

**Formal analysis:** Alebachew Ferede Zegeye, Enyew Getaneh Mekonen, Belayneh Shetie Workneh, Tadesse Tarik Tamir.

**Funding acquisition:** Alebachew Ferede Zegeye.

**Investigation:** Enyew Getaneh Mekonen, Berhan Tekeba, Alemneh Tadesse Kassie.

**Methodology:** Alebachew Ferede Zegeye, Enyew Getaneh Mekonen, Berhan Tekeba, Tewodros Getaneh Alemu, Almaz Tefera Gonete, Alemneh Tadesse Kassie, Mulugeta Wassie.

**Project administration:** Enyew Getaneh Mekonen, Mohammed Seid Ali.

**Resources:** Berhan Tekeba, Tewodros Getaneh Alemu, Mohammed Seid Ali, Belayneh Shetie Workneh, Tadesse Tarik Tamir.

**Software:** Alebachew Ferede Zegeye, Almaz Tefera Gonete, Belayneh Shetie Workneh, Tadesse Tarik Tamir, Mulugeta Wassie.

**Supervision:** Mohammed Seid Ali.

**Validation:** Alebachew Ferede Zegeye, Tewodros Getaneh Alemu, Mohammed Seid Ali, Alemneh Tadesse Kassie, Belayneh Shetie Workneh.

**Visualization:** Tewodros Getaneh Alemu, Mohammed Seid Ali, Almaz Tefera Gonete, Mulugeta Wassie.

**Writing – original draft:** Alebachew Ferede Zegeye, Mohammed Seid Ali, Alemneh Tadesse Kassie.

**Writing – review & editing:** Mulugeta Wassie.

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
