## [Decision Letter · Decision Letter 0]

7 May 2024

PONE-D-24-01758Spatial distribution and determinants of measles vaccination dropout among under-five children in Ethiopia: A spatial and multilevel analysis of 2019 Ethiopian Demographic and Health SurveyPLOS ONE

Dear Dr. Zegeye,

Thank you for submitting your manuscript to PLOS ONE. After careful consideration, we feel that it has merit but does not fully meet PLOS ONE’s publication criteria as it currently stands. Therefore, we invite you to submit a revised version of the manuscript that addresses the points raised during the review process.

We look forward to receiving your revised manuscript.

Kind regards,

Addisalem Workie

Academic Editor

PLOS ONE

"https://bmcpublichealth.biomedcentral.com/counter/pdf/10.1186/s12889-023-15493-2.pdf

https://www.tandfonline.com/doi/full/10.1080/21645515.2022.2035558

https://journals.plos.org/plosone/article?id=10.1371/journal.pone.0284781

https://www.tandfonline.com/doi/full/10.1080/21645515.2022.2035558

https://bmcpublichealth.biomedcentral.com/articles/10.1186/s12889-022-12914-6

In your revision ensure you cite all your sources (including your own works), and quote or rephrase any duplicated text outside the methods section. Further consideration is dependent on these concerns being addressed.

3. Thank you for including your ethics statement:  "N/A".   

1.) For studies reporting research involving human participants, PLOS ONE requires authors to confirm that this specific study was reviewed and approved by an institutional review board (ethics committee) before the study began. Please provide the specific name of the ethics committee/IRB that approved your study, or explain why you did not seek approval in this case.

2.) Please provide additional details regarding participant consent. In the ethics statement in the Methods and online submission information, please ensure that you have specified (1) whether consent was informed and (2) what type you obtained (for instance, written or verbal, and if verbal, how it was documented and witnessed). If your study included minors, state whether you obtained consent from parents or guardians. If the need for consent was waived by the ethics committee, please include this information.

**Comments to the Author**

1. Is the manuscript technically sound, and do the data support the conclusions?

Reviewer #1: Partly

Reviewer #2: Yes

2. Has the statistical analysis been performed appropriately and rigorously? 

Reviewer #1: N/A

Reviewer #2: Yes

3. Have the authors made all data underlying the findings in their manuscript fully available?

Reviewer #1: Yes

Reviewer #2: No

4. Is the manuscript presented in an intelligible fashion and written in standard English?

Reviewer #1: Yes

Reviewer #2: Yes

5. Review Comments to the Author

Reviewer #1: 1. The result section of the abstract should be reconsidered. For example, you say that marital status had higher odds of a measles vaccination dropout rate. Is it the single or married group that has higher odds of a drop rate? This comment also includes the factors.

2. Background section: Paragraph 2, second and third sentences convey similar idea. You should use either of one.

3. Paragraph 4, first sentence is repeated a second time (1-2 million....).

4. Methods: Study setting- latitude should be rewritten.

Study design and period: Your study is a secondary data analysis. however, you dictated as you you conducted the primary data collection. I suggest this section to be rewritten.

Data source and sampling procedure: Binary logistic regression model and multi-level analysis model are used by the authors. How this two models applied simultaneously as the assumptions of the two models are different?

Dependent variables: In determining the dropout rate, the descriptive sentences and the formula are not similar- should be rewritten in away that is clear for the readers.

Reviewer #2: Immunization is the process of administering a vaccination to develop immunity against an infectious agent, with the primary goal being to prevent the disease that the infection causes (3, 4).

Comment: Change “a vaccination” to “a vaccine”. Change “with the primary goal being to prevent the disease that the infection causes” to “with the primary goal of preventing the disease caused by the infection”

Childhood immunization, which prevents 1-2 million childhood deaths worldwide

Comment: annually? specify period

Immunization saves millions of lives annually, making it a success story for global health and development.

Comment: Given the previous sentence, this sentence is unnecessary. I suggest deleting it.

The measles vaccine alone prevented 23 million deaths between 2010 and 2018.

Comment: Based on the following reference: https://www.cdc.gov/mmwr/volumes/68/wr/mm6848a1.htm

the 23 million deaths prevented refer to the 2000-2018 period

More than 116 million infants, or 86% of all babies born, receive vaccinations each year, which is an all-time high (10, 11). In sub-Saharan Africa, 4.4 million children lose their lives due to communicable diseases each year as a result of inadequate vaccination and setup, incomplete vaccination, and impediments to delivering vaccinations, although these deaths may have been avoided by vaccination (12, 13).

Comment: Given the previous sentence, the focus is already on the measles vaccine. Why go back on all vaccines in general? Isn't that complementary to what was written above (“Each year, vaccine-preventable diseases such as … incomplete vaccination coverage (6-8).”)?

Although all children are expected to receive the measles vaccination at nine months, in Ethiopia, as part of the government's subsidized programme, 3.4% of children did not receive complete measles virus immunization through routine immunization services in 2015 (14).

Comment: Please provide more details on measles incidence, case and mortality estimates, and vaccine coverage in the African region, if possible, and in Ethiopia in particular. Examples of how other authors have done this may be found here: https://www.cdc.gov/mmwr/volumes/72/wr/mm7236a3.htm?s_cid=mm7236a3_w.

A measles vaccination dropout occurs when a child has received the first recommended dose of the vaccine but has missed the second recommended measles vaccination dose.

Comment: It would be much more relevant to show the gap between MCV1 and MCV2 before introducing the concept of vaccine dropout.

Vaccination dropout is used as an indicator of immunization program performance, and low dropout rates indicate good access and utilization of immunization services

Comment: Add “to” after good access

If they are more than 10%, there is typically a major quality issue that has to be addressed (15, 18).

Comment: For consistency, this sentence needs to follow: “Vaccination dropout is used as an indicator of immunization program performance, and low dropout rates indicate good access to and utilization of immunization services (17)”.

Despite achievements observed in the reduction of under-five mortality rates, about 1-2 million children are still dying each year due to vaccine-preventable diseases, and sub Saharan African countries share the huge burden of global under-five mortality.

Comment: This point has already been discussed above. What is the point of mentioning it again?

As far as our search of the literature and knowledge goes, the risk (hotspot) areas for measles vaccination dropout among under-five children have not been identified in Ethiopia.

Comment: What about: https://www.mdpi.com/2076-393X/12/3/328 ? How is this study different from yours?

Methods

Data source and sampling procedure

The Ethiopian Demographic and Health Survey (EDHS 2019) data was used to investigate the

spatial distribution of measles vaccination dropouts and determinates among under-five children

Comment: Do you mean determinants among under-five children?

A binary logistic regression model was used to determine the factors associated with measles vaccination dropout rates. Determinants of the measles vaccination dropout were reported in terms of an adjusted odds ratio (AOR) with a significance level of 95%. In the univariable analysis, at 95% confidence intervals, a p-value of <0.25 was considered a candidate for the multivariable analysis of the data. All variables with p-values <0.05 were considered statistically significant in multivariable logistic regression.

Comment: This part should not be in the data source and sampling procedure section

Spatial analysis

Using Getis-OrdGi* statistics, significant “hot-spot” and “colds-pot” areas for measles vaccination dropout have been identified.

Comment: Please provide arguments justifying the use of Getis-OrdGi* statistics instead of local Moran’s I (LISA) statistic measures in your case

Study variables

The measles vaccination dropout rate was calculated by dividing the number of children aged 0-59 months who received measles 1 by the number of children aged 0-59 months who received measles 2 divided by the number of children 0-59 months of age who received measles 1 multiplied by 100%;

Comment: The formula does not match the given operational definition of measles vaccination dropout.

Birth interval (months) (<24, 24-28, >48)

Comment: Do you mean <24, 24-48, >48?

Number of under-5 children (≤1, 2, ≥3)

Comment: ≤1 means 0 to 1. Can you explain how it is possible to drop out of measles vaccination if there are 0 children under 5 years old?

Random effects (Measures of variation)

Moreover, the PCV

Comment: Change “PCV” to “proportional change in variance (PCV)”

Results

Hotspot analysis of measles vaccination dropout

The local Getis-Ord Gi* statistics were employed in the present study to determine the measles vaccination dropout hot and cool spots.

Comment: Change “cool” to “cold”

The odds of measles vaccination dropout rates were 1.87 times higher among under-five children whose birth interval was less than 24 months compared to under-five children whose birth interval was greater than 48 months

Comment: Change “compared to” to “compared with”

Measles vaccination dropout rates were 3.98 times higher among under-five children born from unmarried mothers as compared to under-five children delivered from mothers who had married

Comment: Change to “among children under five born to unmarried mothers than among children under five born to married mothers”

The odds of measles vaccination dropout rates were 3.86 times higher among under five children, where the number of under-five children is one and less compared to the number of under-five children of three or more

Comment: Change “compared to” to “compared with”

I suggest to the authors to merge table 3 and table 4. Examples of how other authors have done this may be found in the following articles:

https://bmcpublichealth.biomedcentral.com/articles/10.1186/s12889-015-2315-z

https://bmcpublichealth.biomedcentral.com/articles/10.1186/s12889-018-5881-z

Discussion

Despite a modest improvement in vaccination rates globally from 2021 to 2022, 33 million children were still not vaccinated against measles: around 22 million missed their first dose, and an additional 11 million, or 74% of the total, missed their second dose (30).

Comment: Please also provide details on children missing measles vaccine doses in Ethiopia.

In this study, the prevalence of measles vaccination dropout among under-five children in Ethiopia was found to be 33.26% (95% CI: 31.65, 34.91).

Comment: Not necessary to add: “(95% CI: 31.65, 34.91)”

The potential explanation for these discrepancies may be due to differences in health policy, quality of health care, and socioeconomic and cultural differences across the countries.

Comment: Please provide references to support your assumptions.

The spatial autocorrelation statistic confirmed that the distribution of measles vaccination dropout was clustered in some geographical areas of Ethiopia (Moran’s I = 0.358552, p <0.001).

Comment: No need to add: “(Moran’s I = 0.358552, p <0.001)”.

The most plausible explanation for this geographical variance in measles vaccine dropout rates could be due to significant differences in infrastructure, such as the distribution of healthcare facilities and professionals among regions, as well as differences in roads, electricity, and water.

Comment: Please provide references to support your assumptions.

The possible explanation might be that short birth intervals can cause a burden on a woman's ability to care for her dependent children as well as other responsibilities in the home. Because of this, mothers might not have the time or desire to bring their children to the health institution to get essential vaccines.

Comment: Please provide references to support your assumptions.

The possible explanation could be that a large family size is likely to strengthen maternal capabilities to extend more care to the younger children as well as her mobility to get access to immunization services due to experience.

Comment: Please provide references to support your assumptions. How can you explain the observed discrepancies between your findings and those of the studies conducted in Ghana, Australia, India, Lebanon, and Brazil?

Measles vaccination dropout was 2.43 times higher among under-five children whose place of residence was rural as compared to under-five children from urban residence. This is consistent with the findings from previous studies in Malawi (55). The possible explanation might be that the low vaccination dropout rate among urban children may be due to the following factors: highly educated mothers may have greater control and autonomy over residence resources; they may also have changed traditional perspectives about immunization. As a result, they might be more likely to seek healthcare and acquire new information about health issues earlier.

Comment: There is substantial evidence that children living in rural areas are more likely to drop out measles vaccination than children living in urban areas. Please provide more references. Provide also references to support your assumptions.

The findings of this study are contradictory to those of the studies conducted in India (59) and Bangladesh (60). The possible explanation for the inconsistency might be Low vaccination dropouts in rural areas are due to the use of traditional birth attendants and primary health care workers, who both play a role in encouraging mothers to attend the maternal and child health clinics, although these roles do not formally exist in urban areas.

Comment: It would be much more relevant to mention other studies conducted in African settings. Provide also references to support your assumptions.

The odds of measles vaccination dropout rates were 3.20 times higher among under-five children who have low community level ANC utilization as compared to under-five children whose community level ANC utilization was high. This finding is in line with other studies in Ethiopia (63) and Gambia (33). This association might be a result of the fact that mothers who do not make full use of ANC services miss out on opportunities to get information about the importance and timing of child vaccinations. In addition, mothers who decide not to receive antenatal care or give birth in a healthcare facility might not give the same attention to childhood vaccinations as their peers from similar socioeconomic backgrounds.

Comment: Please provide references to support your assumptions.

Under-five children whose residence is in the Benishangul Gumuz region have a 1.80 times higher measles vaccination dropout rate compared to the Amhara region. The possible explanation for the association might be that even though vaccinations are free in Ethiopia, mothers must still cover certain indirect costs (such as food and transportation expenses), which frequently restrict their use of maternity and child health services, including immunization. In this sense, mothers in the lower income category are more likely to encounter these obstacles when trying to get to healthcare services, which eventually results in immunization dropouts.

Comment: Please provide references to support your assumptions.

Strength and Limitations of the study

Comment: Strengths and limitations of the study are clearly described. However, strengths of the EDHS may be more detailed and explicit. In addition, more about limitations to generalize information may be more explicit, vast and comprehensive.

Conclusions and recommendations

Comment: Given the relevance of the findings, the conclusions and recommendations are rather laconic.

6. PLOS authors have the option to publish the peer review history of their article (what does this mean?). If published, this will include your full peer review and any attached files.

Reviewer #1: No

Reviewer #2: No

---

## [Author Response · Author response to Decision Letter 0]

16 May 2024

Response to reviewers has been submitted.

---

## [Decision Letter · Decision Letter 1]

28 May 2024

PONE-D-24-01758R1Spatial distribution and determinants of measles vaccination dropout among under-five children in Ethiopia: A spatial and multilevel analysis of 2019 Ethiopian Demographic and Health SurveyPLOS ONE

Dear Dr. Zegeye,

Thank you for submitting your manuscript to PLOS ONE. After careful consideration, we feel that it has merit but does not fully meet PLOS ONE’s publication criteria as it currently stands. Therefore, we invite you to submit a revised version of the manuscript that addresses the points raised during the review process.

We look forward to receiving your revised manuscript.

Kind regards,

Addisalem Workie Demsash

Academic Editor

PLOS ONE

Journal Requirements:

Reviewers' comments:

Reviewer's Responses to Questions

**Comments to the Author**

1. If the authors have adequately addressed your comments raised in a previous round of review and you feel that this manuscript is now acceptable for publication, you may indicate that here to bypass the “Comments to the Author” section, enter your conflict of interest statement in the “Confidential to Editor” section, and submit your "Accept" recommendation.

Reviewer #2: All comments have been addressed

2. Is the manuscript technically sound, and do the data support the conclusions?

Reviewer #2: Yes

3. Has the statistical analysis been performed appropriately and rigorously? 

Reviewer #2: Yes

4. Have the authors made all data underlying the findings in their manuscript fully available?

Reviewer #2: No

5. Is the manuscript presented in an intelligible fashion and written in standard English?

Reviewer #2: Yes

6. Review Comments to the Author

**Reviewer #2:** The authors have answered all my comments. Some minor comments:

Introduction

For greater consistency, the paragraph on “The interval between the first and … community poverty (19-21)” and the one on “A measles vaccination dropout occurs … place of residence (24, 25)” should be merged and rewritten as they are complementary.

Discussion

The potential explanation for these discrepancies may be due to differences in health policy, quality of health care, and socioeconomic and cultural differences across the countries.

Comment: References were provided to support the hypotheses. However, it does not seem sufficient to reflect the studies conducted in Nigeria and Ghana on the one hand and Gambia, South Africa and India on the other. It is necessary to be more explicit to emphasize the factual similarities and differences based on health policy, quality of health care, and socioeconomic and cultural characteristics.

7. PLOS authors have the option to publish the peer review history of their article (what does this mean?). If published, this will include your full peer review and any attached files.

Reviewer #2: No

---

## [Author Response · Author response to Decision Letter 1]

29 May 2024

Response to reviewers has been uploaded

---

## [Editor Report · Decision Letter 2]

30 May 2024

Spatial distribution and determinants of measles vaccination dropout among under-five children in Ethiopia: A spatial and multilevel analysis of 2019 Ethiopian Demographic and Health Survey

PONE-D-24-01758R2

Dear Dr. Zegeye,

We’re pleased to inform you that your manuscript has been judged scientifically suitable for publication and will be formally accepted for publication once it meets all outstanding technical requirements.

Kind regards,

Addisalem Workie Demsash

Academic Editor

PLOS ONE
---

## [Editor Report · Acceptance letter]

27 Jun 2024

PONE-D-24-01758R2 

PLOS ONE

Dear Dr. Zegeye, 

I'm pleased to inform you that your manuscript has been deemed suitable for publication in PLOS ONE. Congratulations! Your manuscript is now being handed over to our production team.

Kind regards, 

on behalf of

Mr. Addisalem Workie Demsash 

Academic Editor

PLOS ONE